# Smoothly Bounding User Contributions in Differential Privacy

**Alessandro Epasto**
Google Research
111 8th Ave,
New York, NY, 10011
aepasto@google.com

**Mohammad Mahdian**
Google Research
111 8th Ave,
New York, NY, 10011
mahdian@google.com

**Jieming Mao**
Google Research
111 8th Ave,
New York, NY, 10011
maojm@google.com

**Vahab Mirrokni**
Google Research
111 8th Ave,
New York, NY, 10011
mirrokni@google.com

**Lijie Ren**
Google Research
111 8th Ave,
New York, NY, 10011
renlijie@google.com

## Abstract

A differentially private algorithm guarantees that the input of a single user won't significantly change the output distribution of the algorithm. When a user contributes more data points, more information can be collected to improve the algorithm's performance. But at the same time, more noise might need to be added to the algorithm in order to keep the algorithm differentially private and this might hurt the algorithm's performance. [AKMV19] initiates the study on bounding user contributions and proposes a very natural algorithm which limits the number of samples each user can contribute by a threshold.

For a better trade-off between utility and privacy guarantee, we propose a method which smoothly bounds user contributions by setting appropriate weights on data points and apply it to estimating the mean/quantiles, linear regression, and empirical risk minimization. We show that our algorithm provably outperforms the sample limiting algorithm. We conclude with experimental evaluations which validate our theoretical results.

## 1 Introduction

The notion of *Differential Privacy*, introduced by [DMNS06], aims to capture the requirement that the output of an algorithm should not reveal much about the information provided by a single user. The classical definition of differential privacy assumes each user controls one row in the input data set, and guarantees that the removal (or change) of one row in the data set does not change the output significantly.

In many applications of differential privacy, a single user might contribute more than one data point. A prominent example, which is the focus of this paper, is private machine learning, where a user often provides several points in the training data set. While the standard definition of differential privacy can still capture such settings by defining a row as the collection of all data points belonging to the same user, an important and useful nuance is lost in this translation. Most importantly, when a user contributes many data points, the algorithm designer must balance between the value of the information contained in these data points, and the added noise it will have to add to the output to make it private with respect to this user.

[AKMV19] initiated the study of this problem, and proposed a natural algorithm which limits the number of samples each user can contribute by a threshold. This threshold is then optimized to strike the right balance between the error due to the noise, and the bias introduced by removing the samples.

This sample limiting algorithm has two drawbacks: (i) It completely discards some data points from users who have too many data points and the information of these data points is lost. (ii) Some data points may contain more useful information than the others but the sample limiting algorithm treats all data points the same when deciding which data points to discard. Our goal in this paper is to answer this question: is it possible to significantly improve over sample limiting by bounding the contribution of each user in a way that is more smooth and careful about the information contained in each sample?

To answer this question, we propose a weighted averaging method to smoothly bound user contributions. The main idea of this method is to set appropriate weights on data points instead of completely discarding some data points.

## 1.1 Our results

In Section 3, as a warm-up, we study a simple problem: estimating the mean. For this problem, finding the optimal algorithm corresponds to finding the right weights when averaging samples. We compute the overall error of the algorithm in terms of these weights, and show how the optimal set of weights can be found. We then compare the error of such an optimal algorithm with that of the best sample limiting approach. We present examples showing that the error of the sample limiting method can be asymptotically 1.5 times higher than that of our algorithm. However, as we prove, this gap cannot exceed 4.

In Section 4, we extend the weighted averaging algorithm to empirical risk minimization by minimizing a weighted version of empirical risk. Our main technical contribution is to prove a weighted version of uniform convergence, which could be of independent interest. We also extend the weighted averaging algorithm to estimating quantiles in a similar way (in Appendix B). Similarly as the warm-up problem, for ERM and estimating quantiles, our weighted averaging algorithm has advantage over the sample limiting algorithm, but the advantage is limited.

In Section 5, we study linear regression with label privacy (defined in Section 5). We show that label privacy allows us to design the weights better based on the usefulness of data points and the weighted averaging algorithm can have a much bigger advantage over the sample limiting algorithm. In particular, we prove that the optimal algorithm can be parameterized by a matrix $C$, and calculate the error as a function of this matrix $C$. Next, we prove that this function is convex, and therefore, one can compute the optimal $C$ in polynomial time. In Section 5.3, we study the gap between the error of our algorithm and the sample limiting algorithm with the best possible threshold, and prove that this gap can be unbounded. In other words, there are instances where our algorithm has an error that is better than the best sample limiting algorithm by an arbitrary factor.

Finally, in Section 6, we analyze the performance of our algorithm and its comparison with sample limiting empirically using some real-world data sets as well as generated data for linear regression with label privacy. This empirical study shows that our algorithms achieve lower loss compared to the baseline methods (e.g., sample limiting) – confirming our theoretical results. We also include in Appendix E experiment results on logistic regression using our ERM algorithm.

## 1.2 Related work

Differential privacy is proposed by the seminal work of [DMNS06]. For a detailed survey on differential privacy, see [DR14].

Differentially private linear regression and its general form, empirical risk minimization have been well-studied [CM08, CMS11, KST12, JKT12, TS13, SCS13, DJW13, JT14, BST14, Ull15, TTZ15, STU17, WLK⁺17, WYX17, ZZMW17, Wan18, She19a, She19b, INS⁺19, BFTT19, WX19, FKT20]. In particular, [WX19] studies label privacy which is similar to the setting we have in Section 5. These results are in the case when each user has only one data point.

Motivated by federated learning, [AKMV19] initiates the study of bounding user contributions in differential privacy. [TAM19, PSY⁺19] study how to adaptively bound user contributions in

differentially private stochastic gradient descent for federated learning. For a detailed survey on federated learning, see [KMA+19]. More broadly, our setting of each user having multiple data points can be considered as a special case of personalized/heterogeneous differential privacy [JYC15, SCS15, AGK17] and is very related to group privacy which is introduced in [Dwo06].

## 2 Preliminaries

### 2.1 Differential privacy

We define the notion of *differential privacy* for an algorithm $\mathcal{A}$ that takes as input a data set $D$ from the space $\mathcal{D}$ of all possible data sets, and produces an output $\mathcal{A}(D)$ in the space of outputs $\mathcal{O}$. Typically, $D$ is a collection of $n$ data points for some $n$. To define differential privacy, we need a notion of *neighboring* data sets. In the classical setting of differential privacy, two data sets $D, D' \in \mathcal{D}$ are called neighboring data sets, denoted $D \sim D'$, if one is obtained from the other by removing one data point. In Section 2.2, we will discuss a more general notion that captures settings where a user controls more than one data point.

**Definition 1** (Differential Privacy [DMNS06]). *A randomized algorithm $\mathcal{A}$ is $(\varepsilon, \delta)$-differentially private ($(\varepsilon, \delta)$-DP for short) if for all neighboring data sets $D, D' \in \mathcal{D}$, and all subsets of outcomes $S \subseteq \mathcal{O}$,*

$$\Pr[\mathcal{A}(D) \in S] \leq e^{\varepsilon} \Pr[\mathcal{A}(D') \in S] + \delta.$$

When $\delta = 0$, we say that $\mathcal{A}$ is $\varepsilon$-DP.

The Laplace mechanism [DMNS06] is a standard technique to achieve differential privacy by adding Laplace noise of appropriate scale to the outcome of computation.

**Definition 2** ($\ell_1$-sensitivity). *The $\ell_1$-sensitivity of a function $f : \mathcal{D} \rightarrow \mathbb{R}^d$ is: $\Delta f = \max_{D \sim D'} \|f(D) - f(D')\|_1$.*

**Definition 3** (Laplace Mechanism [DMNS06]). *Given any function $f : \mathcal{D} \rightarrow \mathbb{R}^d$, the Laplace mechanism is defined as $f(D) + (W_1, ..., W_d)$ where $W_i$'s are i.i.d random variables drawn from $Lap(\Delta f / \varepsilon)$. Here $Lap(b)$ is the Laplace distribution with mean $0$ and variance $2b^2$.*

**Theorem 1** ([DMNS06]). *The Laplace mechanism is $\varepsilon$-DP.*

### 2.2 User-level differential privacy

In this paper, we consider the setting in which there are $m$ users owning $n$ data points and $m \leq n$. Therefore, a single user can have more than one data point. For each user $l \in [m]$, we use $S_l$ to denote the set of indices of data points owned by user $l$, and let $s_l = |S_l|$. We assume $S_l$'s are publicly known. We focus on the case when user data points are sampled from the same distribution. When user data distributions are heterogeneous, additional bias need to be deal with (as in [AKMV19]), and we do not consider this case.

The user-level differential privacy definition mostly follows Definition 1. The only difference is that now two data sets $D, D'$ are considered to be neighboring data sets if they are the same except all data points from a single user $l$.

## 3 Warm-up: estimating the mean

For warm-up, we consider a simple setting where we have $n$ data points $y_1, ..., y_n$ generated as $y_i = \beta + \xi_i$ and we want to estimate the unknown mean $\beta$ differentially privately. Here noise $\xi_i$'s are independent with mean $0$ and variance $\sigma^2$. We assume all $y_i$'s are bounded, i.e., $y_i \in [0, B]$.

We want to minimize the expected squared error of our estimate $\tilde{\beta}$: $\mathbb{E}[(\beta - \tilde{\beta})^2]$. This is just the variance of $\tilde{\beta}$ when $\beta$ is unbiased ($\mathbb{E}[\tilde{\beta}] = \beta$). The expectation is over the randomness of our algorithm and $\xi_i$'s.

## 3.1 The weighted averaging algorithm

In this subsection, we propose our weighted averaging algorithm $WA_c$ which is parameterized by non-negative weights $c_1, ..., c_n$ with $\sum_i c_i = 1$. This algorithm simply computes the weighted average of the input $y_i$'s, and applies the Laplace mechanism to this average:

---

**Algorithm 1** Weighted Averaging $WA_c$

---

**Input:** $y_1, \ldots, y_n \in \mathbb{R}$
**Parameters:** $c_1, \ldots, c_n \in \mathbb{R}_{\geq 0}$, with $\sum_{i=1}^n c_i = 1$
  1: $\hat{\beta} \leftarrow c_1 y_1 + \cdots + c_n y_n$
  2: $\tilde{\beta} \leftarrow \hat{\beta} + \text{Lap}\left(\frac{B \cdot \max_{l=1}^m \sum_{i \in S_l} c_i}{\varepsilon}\right)$
  3: **return** $\tilde{\beta}$

---

In the following two lemmas, we prove that $WA_c$ is $\varepsilon$-DP and analyze its variance.

**Lemma 1.** *For every c, the algorithm $WA_c$ is $\varepsilon$-DP.*

*Proof.* If some user $l$ change its input $y_i$'s for $i \in S_l$, $\hat{\beta}$ would be changed additively by at most $B \cdot \sum_{i \in S_l} c_i$. Therefore, the $\ell_1$-sensitivity of $\hat{\beta}$ is $B \cdot \max_l \sum_{i \in S_l} c_i$. By Theorem 1, we know that $\tilde{\beta}$ is $\varepsilon$-DP. $\square$

It is easy to check that $\tilde{\beta}$ is unbiased. We will just analyze its variance.

**Lemma 2.** *For every c, the variance of the output of $WA_c$ can be written as:* $\text{Var}\left(\tilde{\beta}\right) = \sigma^2(c_1^2 + \cdots + c_n^2) + 2\left(\frac{B \cdot \max_{l=1}^m \sum_{i \in S_l} c_i}{\varepsilon}\right)^2.$

*Proof.* For the variance of $\tilde{\beta}$, since $c_1 y_1, ..., c_n y_n$ and the Laplace noise are independent, we have

$$\text{Var}\left(\tilde{\beta}\right) = \sum_{i=1}^n \text{Var}(c_i y_i) + \text{Var}\left(\text{Lap}\left(\frac{B \cdot \max_{l=1}^m \sum_{i \in S_l} c_i}{\varepsilon}\right)\right)$$

$$= \sigma^2(c_1^2 + \cdots + c_n^2) + 2\left(\frac{B \cdot \max_{l=1}^m \sum_{i \in S_l} c_i}{\varepsilon}\right)^2.$$

$\square$

Next, we characterize the weight vector $c$ that minimizes $\text{Var}(\tilde{\beta})$. The proof of Lemma 3 can be found in the supplementary material.

**Lemma 3.** *Let $c^* = (c_1^*, ..., c_n^*)$ be the vector that minimizes $\text{Var}(\tilde{\beta})$. There exists h, such that,*

    *1. $s_1 \leq h \leq s_m$*

    *2. For each data point $i$ of user $q$, $c_i^* = \frac{\min(h, s_q)}{s_q \sum_{l=1}^m \min(h, s_l)}$.*

Define $n_h = \sum_{l=1}^m \min(h, s_l)$. Using the characterization we get in Lemma 3, we show in the next claim that minimizing $\text{Var}(\tilde{\beta})$ can be simplified into minimizing a function of a single variable $h$.

**Claim 1.** *For the weighted averaging algorithm, the minimum of $\text{Var}(\tilde{\beta})$ equals to* $\min_{h: s_1 \leq h \leq s_m} v(h)$, *where* $v(h) = \sigma^2 \sum_{l=1}^m s_l \cdot \left(\frac{\min(h, s_l)}{n_h \cdot s_l}\right)^2 + 2\left(\frac{B \cdot h}{\varepsilon \cdot n_h}\right)^2.$

*Proof.* For any $h > 0$, when setting $c_i = \frac{\min(h, s_q)}{s_q \sum_{l=1}^m \min(h, s_l)}$ for any data point $i$ of any user $q$, it is easy to check that $\text{Var}(\tilde{\beta}) = v(h)$. Then by Lemma 3, we get the claim. $\square$

Regarding weights computation, Lemma 3 and Claim 1 show that finding the optimal weight vector $c$ is equivalent to minimizing a function $v(h)$ of a single parameter $h \in [s_1, s_m]$. Optimizing this function of a single parameter can be simply done by setting the derivative to be 0 and considering locations where the derivative is not continuous.

## 3.2 The sample limiting algorithm

The sample limiting algorithm picks an integer threshold $h$ between $s_1$ and $s_m$. For each user $l$, the sample limiting algorithm arbitrarily selects $\min(s_l, h)$ data points and apply the Laplace mechanism to the average of $n_h = \sum_{l=1}^m \min(h, s_l)$ selected data points. In other words, if we let $T$ denote the set of selected samples, the sample limiting algorithm outputs $\tilde{\beta} = \frac{1}{n_h} \sum_{i \in T} y_i + \mathrm{Lap}\left(\frac{B \cdot h}{\varepsilon \cdot n_h}\right)$.

It is easy to see that the sample limiting algorithm is a special case of the weighted averaging algorithm with weights $c_i = \mathbb{1}[i \in T]/n_h$. Therefore, by Lemma 1, we know the output of the sample limiting algorithm is $\varepsilon$-DP. By Lemma 2, the variance of $\tilde{\beta}$ can be written as follows.

**Claim 2.** *For the sample limiting algorithm with integer threshold $h$, Var($\tilde{\beta}$) can be written as the following function:* $v'(h) = \frac{\sigma^2}{n_h} + 2\left(\frac{B \cdot h}{\varepsilon \cdot n_h}\right)^2$.

## 3.3 Comparing the variances

In this subsection, we compare the minimum variances of two algorithms we describe earlier. By Claims 1 and 2, we just need to compare $\min_{h:s_1 \le h \le s_m} v(h)$ and $\min_{h:s_1 \le h \le s_m, h \in \mathbb{N}} v'(h)$.

Since the sample limiting algorithm is a special case of the weighted averaging algorithm, we know its variance is greater than or equal to that of the best weighted averaging algorithm. We now show that there are examples where the variance of the sample limiting algorithm is larger than the variance of the weighted averaging algorithm by a factor that asymptotically converges to $3/2$.

**Theorem 2.** *For every $g \in \mathbb{N}$, there is an instance where the variance of the best weighted averaging algorithm is less than $\frac{2g+1}{4g^2}$, while the variance of the best sample limiting algorithm is at least $\frac{3}{4g}$. For every $g$, $\frac{3}{4g} \ge \frac{2g+1}{4g^2}$, and the ratio between these two numbers converges to $3/2$ as $g$ goes to infinity.*

On the other hand, we show that the gap between the two minimum variances is at most a factor of 4.

**Theorem 3.** *In every instance, the variance of the best sample limiting algorithm is at most 4 times the variance of the best weighted averaging algorithm.*

# 4 Extension to empirical risk minimization

In this section, we extend the weighted averaging algorithm to empirical risk minimization. Missing proofs and similar extension to estimating quantiles can be found in the supplementary material.

Here we give the setting of empirical risk minimization (ERM). We are given $n$ data points $D = (X_1, ..., X_n)$ from a universe $\mathcal{X}$. They are sampled independently from an unknown distribution $\mu$. We need to optimize over a closed, convex set $\mathcal{C}$ bounded by $B$ (i.e. for all $\theta \in \mathcal{C}$, $\|\theta\|_2 \le B$) and we are given a loss function $l$. For each data point $X \in \mathcal{X}$, $l(\cdot, X)$ defines a loss function on $\mathcal{C}$. We assume $l(\cdot, X)$ is convex and $L$-Lipschitz. Our goal is to minimize the population risk $L_\mu(\theta) = \mathbb{E}_{X \sim \mu}[l(\theta, X)]$ over $\theta \in \mathcal{C}$ and we define $\theta^* \in \mathcal{C}$ to be the optimal solution: $\theta^* \in \arg\min_{\theta \in \mathcal{C}} L_\mu(\theta)$.

Now we describe our weighted ERM algorithm (Algorithm 2) parametrized by non-negative weights $c = (c_1, ..., c_n)$. The main idea is to consider the weighted empirical risk $\hat{L}(\theta, c, D) = \sum_{i=1}^n c_i l(\theta, X_i)$ for dataset $D = (X_1, ..., X_n)$. In order to apply standard (record-level) differentially private ERM algorithms and ensure differential privacy with respect to each user, we define a new loss function $l'$. Let $M_j$ to be the meta-data of user $j$: $M_j = (S_j, \{c_i, X_i\}_{i \in S_j})$. We define $l'(\theta, M_j) = \sum_{i \in S_j} c_i l(\theta, X_i)$ (i.e. weighted empirical loss of each user). In general, we can use any DP ERM algorithms for the new loss $l'$. For concreteness, we use Algorithm 1 of [BST14] which achieves nearly optimal empirical risk bound.

---
**Algorithm 2** Weighted ERM
---
**Input:** $X_1, ..., X_n \in \mathcal{X}$, loss function $l$,
**Parameters:** $c_1, \ldots, c_n \in \mathbb{R}_{\geq 0}$, with $\sum_{i=1}^n c_i = 1$
  1: Define user-level weighted loss function $l'$ as stated in the above paragraph
  2: Run $(\varepsilon, \delta)$-DP-ERM algorithm (Algorithm 1 of [BST14]) over user-level loss $l'$ and $m$ users, and obtain its output $\tilde{\theta}$
  3: **return** $\tilde{\theta}$
---

**Theorem 4.** *Algorithm 2 is $(\varepsilon, \delta)$-DP and $\mathbb{E}_{D \sim \mu^n, alg}[L_\mu(\tilde{\theta})] - L_\mu(\theta^*)$ is bounded by*

$$O\left( LB\sqrt{d} \cdot \sqrt{\frac{\log^4(m/\delta)\left(\max_{j=1}^m \sum_{i \in S_j} c_i\right)^2}{\varepsilon^2} + \log(d)\log(n)\sum_{i=1}^n c_i^2} \right).$$

To prove Theorem 4, we prove a weighted version of the uniform convergence result (Theorem 5 of [SSSS09]) to give an upper bound on the generalization error of our weighted ERM algorithm and this result might be of independent interest.

**Theorem 5** (Weighted uniform convergence). *For any $\gamma > 0$ and non-negative weights $c = (c_1, ..., c_n)$ with $\sum_{i=1}^n c_i = 1$, with probability at least $1 - \gamma$ over $D \sim \mu^n$, we have*

$$\sup_{\theta \in \mathcal{C}} |\hat{L}(\theta, c, D) - L_\mu(\theta)| \leq O\left( LB\sqrt{d \log(d/\gamma)\log(n)\sum_{i=1}^n c_i^2} \right).$$

**Tradeoff weights.**   In ERM and also estimating quantiles (details in the supplementary material), for optimizing the algorithm performance, we need to pick weights to minimize a formula in the form of $\left(\max_{j=1}^m \sum_{i \in S_j} c_i\right)^2 + A \cdot \sum_{i=1}^n c_i^2$, where $A$ depends on parameters of the problem (for example, in Theorem 4, by simply moving terms around, $A$ would be $\Theta(\log(d)\log(n)\varepsilon^2/\log^4(m/\delta))$). This general form intuitively explains the tradeoff we need to make: $A \cdot \sum_{i=1}^n c_i^2$ measures how hard it is to optimize the weighted objective with a user-level private algorithm and $\left(\max_{j=1}^m \sum_{i \in S_j} c_i\right)^2$ measures how well the weighted objective generalizes.

It is not hard to see that $\text{Var}\left(\tilde{\beta}\right)$ in Section 3 is also in this form, and the results about characterizing the minimizer and comparisons to the sample limiting algorithm also apply here. And we can also use the same method to compute weights $c = (c_1, ..., c_n)$.

# 5   Linear regression with label privacy

In this section, we consider linear regression. We are given $n$ data points of the form $(X_i, y_i)$, where $X_i \in \mathbb{R}^d$ and $y_i \in [0, B]$ for $i = 1, \ldots, n$. The $y_i$ values are generated as $y_i = \beta \cdot X_i + \xi_i$, for a vector $\beta \in \mathbb{R}^d$ unknown to us, and random variables $\xi_i$ representing noise. These random variables are assumed to be independent, each with a mean of $0$ and a variance of $\sigma^2$. We provide a detailed preliminaries to linear regression in the supplementary matirel.

We focus on label-privacy (introduced in [CH11]) in which we protect the privacy of label $y_i$'s and $X_i$'s are public data. In other words, $D$ and $D'$ are considered to be neighboring databases if they have the same $X_i$'s and their $y_i$'s are also the same except data points from just one user $l$.

Our goal is to have an $\varepsilon$-DP algorithm which estimate $\beta$ by outputting an unbiased estimator $\tilde{\beta}$ and minimizes the squared error $\mathbb{E}\left[\sum_{j=1}^d (\beta_j - \tilde{\beta}_j)^2\right]$. Since $\tilde{\beta}$ is unbiased, minimizing the squared error is equivalent to minimizing the variance $\sum_{j=1}^d \text{Var}(\tilde{\beta}_j)$.

## 5.1 The Algorithm

The weighted averaging algorithm of Section 3.1 generalizes the simple averaging algorithm and is a generic linear unbiased estimator for the simple problem. Similarly, for linear regression, we need a generalization of the OLS (ordinary least squared) estimator that provides a generic linear unbiased estimator for $\beta$. Such a generalization has already been proposed by [Ait34], albeit for a different purpose[1].

As a fact (see the supplementary material for details), any linear unbiased estimator can be written as $\hat{\beta} = Cy$, for a $d \times n$ matrix $C = [c_{i,j}]_{d \times n}$ satisfying $CX = I_d$. Our generalization of the weighted averaging algorithm to the higher-dimensional case is to use such an estimator followed by the Laplace mechanism:

---
**Algorithm 3** Generalized Weighted Averaging $GWA_C$

---
**Input:** $X \in \mathbb{R}^{n \times d}, y \in \mathbb{R}^{n \times 1}$
**Parameters:** $C \in \mathbb{R}^{d \times n}$ satisfying $CX = I_d$
  1: $\hat{\beta} \leftarrow Cy$
  2: $b \leftarrow B\varepsilon^{-1} \cdot \max_l \sum_{j=1}^{d} \sum_{i \in S_l} |c_{j,i}|$
  3: independently draw values $W_1, \ldots, W_d$ from $\text{Lap}(b)$
  4: $\tilde{\beta} \leftarrow \hat{\beta} + (W_1, \ldots, W_d)$
  5: **return** $\tilde{\beta}$

---

In the following theorem, we provide the performance of our algorithm and its proof can be found in the supplementary material.

**Theorem 6.** *For every $d \times n$ matrix $C$ satisfying $CX = I_d$, the algorithm $GWA_C$ is $\varepsilon$-DP, and the total variance $\sum_{j=1}^{d} Var(\tilde{\beta}_j)$ of the vector $\tilde{\beta}$ produced by the algorithm $GWA_C$ can be written as:*

$$\sigma^2 \sum_{j=1}^{d} \sum_{i=1}^{n} c_{j,i}^2 + 2d \left( B\varepsilon^{-1} \cdot \max_l \sum_{j=1}^{d} \sum_{i \in S_l} |c_{j,i}| \right)^2 .$$

*This total variance is a convex function and can be minimized in polynomial time.*

## 5.2 The sample limiting algorithm

We generalize the sample limiting algorithm in Section 3.2 to higher dimensions. The sample limiting algorithm is parameterized by an integer threshold $h$. For each user $l$, it randomly picks $\min(h, s_l)$ data points from the user. Let $U$ be the features and $v$ be the labels of the sample data points. The sample limiting algorithm first computes the OLS of the sample data $\hat{\beta}^s = (U^T U)^{-1} U^T v$. Let $C^s = (U^T U)^{-1} U^T$. For each user $l \in [m]$, let $S_l^*$ be the set of corresponding row numbers in $U$ and $v$. The algorithm finally outputs a vector $\tilde{\beta}^s$ obtained by adding to each entry of $\hat{\beta}^s$ a value drawn i.i.d. from $\text{Lap}(b)$, for $b = B\varepsilon^{-1} \cdot \max_{l=1}^{m} \sum_{j=1}^{d} \sum_{i \in S_l^s} |c_{j,i}^s|$.

Similarly to Section 3.2, after fixing the selected points, the sample limiting algorithm can be considered as a special case of the $GWA_C$ algorithm if we expand $C^s$ to $n$ rows. Therefore the output of the sample limiting algorithm is $\varepsilon$-DP and $\sum_{j=1}^{d} Var(\tilde{\beta}_j^s)$ is

$$\sigma^2 \sum_{j=1}^{d} \sum_{i=1}^{n} c_{j,i}^s{}^2 + 2d \left( B\varepsilon^{-1} \cdot \max_{l=1}^{m} \sum_{j=1}^{d} \sum_{i \in S_l} |c_{j,i}^s| \right)^2 .$$

## 5.3 An unbounded gap

Here we give an example which shows that the optimal GWA algorithm can have a much smaller variance than the sample limiting algorithm. This advantage of GWA algorithm can also be seen in the experiments mentioned in Section 6.

In this example (Example 1), data points are from two orthogonal directions (i.e. $X_i$'s have either $X_{i,1} = 0$ or $X_{i,2} = 0$). To control the user contributions, the sample limiting algorithm wants to pick $h$ big for the first dimension and to pick $h$ small for the second dimension. The sample limiting algorithm has to pick the same threshold $h$ for both dimensions. Intuitively, it cannot avoid big user contributions. A formal proof is provided in the supplementary material.

**Example 1.** *$d = 2$. Set $\sigma = 0$ and $2d(B/\varepsilon)^2 = 1$. Let $g$ be some integer parameter and set the number of users to be $m = 2g^2 + 2$. Now consider the data points of users:*

- *User 1 has 1 data point with $X_i = (g, 0)$.*

- *Each of user $2, ..., g^2 + 1$ has $g$ data points with same $X_i = (1, 0)$*

- *User $g^2 + 2$ has $g$ data points with same $X_i = (0, 1)$.*

- *Each of user $g^2 + 3, ..., 2g^2 + 2$ has 1 data point with $X_i = (0, 1)$*

**Claim 3.** *In Example 1, the minimum variance of the sample limiting algorithm is at least $1/4g^3$ and the minimum variance of the generalized weighted averaging algorithm is at most $1/g^4$.*

Example 1 contained one data point whose norm of $X_i$ is much larger than the others. We provide a different example in the supplementary material to show that a gap still exists even when $|X_i| = 1$ for all data points.

# 6 Experiments

In this section we perform an empirical evaluation of our algorithm and we compare it with the sample limiting algorithm for linear regression in the label-privacy case. In Appendix E, we also provide experimental results on logistic regression using our ERM algorithm of Setion 4.

**Datasets** We evaluated all methods on two *publicly-available* datasets containing real-world data as well as synthetic datasets with ground-truth generated with standard open-source libraries. We stress that no private data has been used in these experiments. We only briefly describe our and experimental setup here, more details are available in Appendix E.

**Synthetic data:** We generated regression problem instances with `sklearn`'s `make_regression` ($n \in [600, 3000]$ samples, $d = 10$ features, bias=0.0 and noise=20). To model user contributions we used the Zipf's (power law) distribution for the number of rows of a user (users contributions are often heavy tailed [AH02]). **Real-world datasets:** We used also two UCI Machine Learning Datasets. **drugs** [GKMZ18] ($n = 3107$, $d = 8$, $m = 502$ users with min 1 and max 63 samples) and **news** [MT18] ($n = 3452$, $d = 10$, $m = 297$ users with min 1 and max 878 samples).

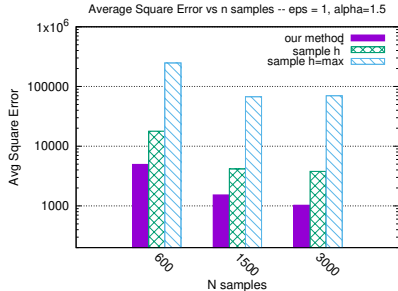

Figure 1: Average squared errors—synthetic dataset, $\alpha = 1.5$ and $\varepsilon = 1$.

**Experimental set up** Experiments are repeated 10 times and we report mean of each metric computed. For quality we use the average squared error for the prediction. We evaluate our general setting algorithm in Section 5 using $\varepsilon = 1, 2, 3$ values. For $\sigma^2$, we treat it as public knowledge and compute it with OLS regression. For **sample limiting** we evaluate the best threshold $h^*$ and the whole datasets (i.e. $h$ set to max user contribution).

**Results on the synthetic dataset** We compare all methods on datasets with varying numbers of samples $n$ and different parameter $\alpha$ of the Zipf's distribution. Lower $\alpha$ values correspond to more uneven distributions (i.e. some users may many more data points than others). The results for $\alpha = 1.5$, $\varepsilon = 1$, are plot in Figures 1. As expected, the larger the number of samples the lower the loss of all methods, however in every setting our method has always significantly lower

| Dataset | $\varepsilon$ | Our method | Sample limit $h^*$ | Sample limit $h_{max}$ |
|---------|---|-----------|------------------|--------------------|
| drugs | 1 | 3.1 | 24.8 | 95.4 |
|       | 2 | 2.5 | 7.7 | 25.2 |
|       | 3 | 2.3 | 4.5 | 12.4 |
| news | 1 | 1696.3 | 96344.4 | 4862670.7 |
|      | 2 | 440.1 | 24110.0 | 1201568.9 |
|      | 3 | 166.2 | 10648.3 | 550989.2 |

Table 1: Average squared errors for our method, sample limit with best threshold ($h^*$), and using all data ($h_max$).

squared error, even orders of magnitude lower (notice the $y$-axis is in log scale). We now fix $\varepsilon = 1$ and $n = 3000$ samples and analyze the effect of the parameter $\alpha$ in Figure 2. Recall that $\alpha$ controls the inequality in the distribution of the user's contributions. As expected, our method is comparatively much better for low $\alpha$ (i.e., more unequal distributions of user contributions), but it performs always better.

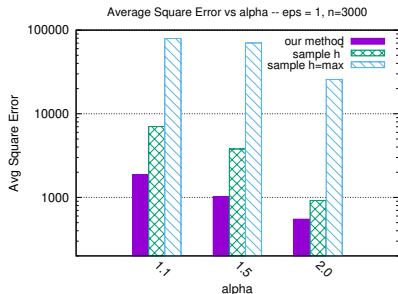

Figure 2: Average squared errors—Synthetic dataset, $\alpha = 1.5, \varepsilon = 1, n = 3000$.

**Results on real-world datasets** We now report the results for the real-world datasets. Our results are summarized in Table 1. The results confirm all empirical observations on the synthetic datasets: the loss decreases for increasing $\varepsilon$ for all methods, but our method has always significantly lower loss than both sample limiting with best and max threshold. Notice that the squared error is overall larger for news than for drugs, this is explained by the larger range of the $y$'s values (in news the values are in $[0, 71]$ vs $[1, 10]$ for drugs).

## 7   Conclusion

In this paper, we propose the weighted averaging method for smoothly bounding user contribution in differential privacy. We apply this method to estimating the mean and quantiles, empirical risk minimization, and linear regression. We show it has advantage over the sampling limiting algorithm, especially in the label-privacy case.

## Broader Impact

Privacy is a fundamental concern in machine learning. Respecting the privacy of the users is a requirement of any real system and differential privacy allows to formalize such requirement. In this paper we provided algorithms with improved trade-offs of utility vs differential privacy. This may enable better outcomes for the users of a system at the same level of privacy. We stress that privacy is only one of the requirements of a real system. Any machine learning technology must also responsibly ensure utility of the system and fairness of the system to the users. Privacy requirements may negatively affect utility, and it is known that differential privacy potentially disparately impacts certain users [BPS19]. Such considerations are beyond the scope of the paper and we refer to the emerging literature on responsible machine learning for addressing them [KR19].

## Funding Transparency Statement

No third-party funding has been used for this research.

## Footnotes

[1][Ait34] proposed the Generalized Least Squares method to solve linear regression when the noise in different observations are correlated.

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
