[Supplementary Material]

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

[2]This captures the expected value for the square of the difference between the true value at $\lambda$ (which is $\beta \cdot \lambda$) and the estimated value at $\lambda$ (which is $\hat{\beta} \cdot \lambda$).

[3] https://archive.ics.uci.edu/ml/datasets.php

[4] https://archive.ics.uci.edu/ml/datasets/Drug+Review+Dataset+%28Druglib.com%29

[5] https://archive.ics.uci.edu/ml/datasets/News+Popularity+in+Multiple+Social+Media+Platforms

[6]`https://www.cvxpy.org/tutorial/advanced/index.html`

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

# A Missing proofs of Section 3

**Lemma 4** (Restatement of Lemma 3). *Let $c^* = (c_1^*, ..., c_n^*)$ be the vector that minimizes $Var(\tilde{\beta})$. There exists $h$, such that,*

1. *$s_1 \leq h \leq s_m$*

2. *For each data point $i$ of user $q$, $c_i^* = \frac{\min(h, s_q)}{s_q \sum_{l=1}^{m} \min(h, s_l)}$.*

*Proof.* We show in this proof how to pick the $h$ as stated in the lemma. Let $M$ be the set $\arg\max_{l=1}^{m} \sum_{i \in S_l} c_i^*$. We start by considering any two data points: data point $i$ of user $p$ and data point $i'$ from of $q$. Consider the following cases:

- $q \in M$ and $p \notin M$: In this case, $c_i^* \leq c_{i'}^*$. If not, pick $\eta = \min((c_{i'}^* - c_i^*)/2, (\max_l \sum_{i \in S_l} c_i^*) - \sum_{i \in S_p} c_i^*)$. Changing $c_i^*$ and $c_{i'}^*$ to $c_i^* + \eta$ and $c_{i'}^* - \eta$ would decrease $\sigma^2((c_1^*)^2 + \cdots + (c_n^*)^2)$ and would not change $\max_l \sum_{i \in S_l} c_i^*$. Therefore this change would decrease $Var(\tilde{\beta})$ and result in a contradiction.

- $p \neq q$ and $p, q \notin M$: In this case $c_i^* = c_{i'}^*$. If not, without loss of generality, assume $c_i^* < c_{i'}^*$. Pick $\eta = \min((c_{i'}^* - c_i^*)/2, (\max_l \sum_{i \in S_l} c_i^*) - \sum_{i \in S_p} c_i^*)$. Similarly as the previous case, changing $c_i^*$ and $c_{i'}^*$ to $c_i^* + \eta$ and $c_{i'}^* - \eta$ would decrease $Var(\tilde{\beta})$ and result in a contradiction.

- $p = q$: In this case $c_i^* = c_{i'}^*$. If not, changing both $c_i^*$ and $c_{i'}^*$ to $(c_i^* + c_{i'}^*)/2$ would decrease $Var(\tilde{\beta})$ and result in a contradiction.

With the above characterization, we are ready to pick $h$. We start with a special case in which $M = [m]$. In this case, $\sum_{i \in S_l} c_i^*$ are the same for all users. Therefore $\sum_{i \in S_l} c_i^* = 1/m$ for any $l$. We also know from the above that data points of the same user have the same $c_i^*$. Therefore, data point $i$ of user $q$ has $c_i^* = \frac{1}{m \cdot s_l}$. Setting $h = s_1$ would work, since

$$\frac{\min(h, s_q)}{s_q \sum_{l=1}^{m} \min(h, s_l)} = \frac{h}{s_l \cdot m \cdot h} = \frac{1}{m \cdot s_l} = c_i^*.$$

Now consider the case in which $M \neq [m]$. Pick an arbitrary user $p \notin M$ and some $i_p \in S_p$. Set $h = (\max_{l=1}^{m} \sum_{i \in S_l} c_i^*)/c_{i_p}^*$. We are going to show such $h$ works as the lemma statement.

For any user $q \in M$ and for any $i \in S_q$, using the above characterization, we know $c_i^* \leq c_{i_p}^*$. Therefore

$$h = \frac{\max_{l=1}^{m} \sum_{i \in S_l} c_i^*}{c_{i_p}^*} = \frac{\sum_{i \in S_q} c_i^*}{c_{i_p}^*} \leq s_q.$$

For any user $q \notin M$ and for any $i \in S_q$, using the above characterization, we know $c_i^* = c_{i_p}^*$. Therefore

$$h = \frac{\max_{l=1}^{m} \sum_{i \in S_l} c_i^*}{c_{i_p}^*} > \frac{\sum_{i \in S_q} c_i^*}{c_{i_p}^*} = s_q.$$

So we have $\sum_{l=1}^{m} \min(h, s_l) = |M|h + \sum_{l \notin M} s_l$. Using the above characterization of two data points, we have

$$\left(|M|h + \sum_{l \notin M} s_l\right) \cdot c_{i_p}^* = \left(\sum_{l \in M} \sum_{i \in S_l} c_i^*\right) + \left(\sum_{l \notin M} \sum_{i \in S_l} c_i^*\right)$$
$$= 1.$$

For any user $q \in M$ and for any $i \in S_q$,

$$\frac{\min(h, s_q)}{s_q \sum_{l=1}^{m} \min(h, s_l)} = \frac{h}{s_q(|M|h + \sum_{l \notin M} s_l)}$$
$$= \frac{h \cdot c_{i_p}^*}{s_q} = \frac{\sum_{i' \in S_q} c_i^*}{s_q} = c_i^*.$$

For any user $q \notin M$ and for any $i \in S_q$,

$$\frac{\min(h, s_q)}{s_q \sum_{l=1}^{m} \min(h, s_l)} = \frac{1}{\sum_{l=1}^{m} \min(h, s_l)} = c_{i_p}^* = c_i^*.$$

$\square$

**Theorem 7** (Restatement of Theorem 2). *For every $g \in \mathbb{N}$, there is an instance where the variance of the best weighted averaging algorithm is less than $\frac{2g+1}{4g^2}$, while the variance of the best sample limiting algorithm is at least $\frac{3}{4g}$. For every $g$, $\frac{3}{4g} \geq \frac{2g+1}{4g^2}$, and the ratio between these two numbers converges to $3/2$ as $g$ goes to infinity.*

*Proof.* We use the following example:

**Example 2.** *For some $g \in \mathbb{N}$, set $m = 2g$. Set $s_l = 1$ for $l = 1, ..., g$ and $s_l = g$ for $l = g+1, ..., 2g$. Set $\sigma = 1$ and $\frac{B}{\varepsilon} = \sqrt{g/2}$.*

The variance of the weighted averaging algorithm can be bounded by:

$$\min_{h: s_1 \leq h \leq s_m} v(h) \leq v(1) = \frac{2g+1}{4g^2}.$$

For sample limiting, for every $h \geq 1$, we have

$$v'(h) = \frac{1}{g + g \cdot h} + g\left(\frac{h}{g + g \cdot h}\right)^2 = g\left(\frac{h}{g + g \cdot h} - \frac{1}{2g}\right)^2 - \frac{1}{4g} + \frac{1}{g} \geq \frac{3}{4g}.$$

Therefore, the variance of the best sample limiting algorithm is $\min_{h: s_1 \leq h \leq s_m, h \in \mathbb{N}} v'(h) \geq \frac{3}{4g}$. $\square$

**Theorem 8** (Restatement of Theorem 3). *In every instance, the variance of the best sample limiting algorithm is at most 4 times the variance of the best weighted averaging algorithm.*

*Proof.* We need to prove $\min_{h: s_1 \leq h \leq s_m, h \in \mathbb{N}} v'(h) \leq 4 \min_{h: s_1 \leq h \leq s_m} v(h)$. Let $h^* = \arg\min_{h: s_l \leq h \leq s_m} v(h)$. Let $p$ be some index such that $s_p \leq h^* \leq s_{p+1}$. Let $q = m - p$. There are at least $q$ users with $s_l \geq h^*$. Set $A = \sum_{l=1}^{p} s_l$. Set $\alpha = A/n_{h^*}$. We consider two cases:

**Case 1 ($\alpha \geq 1/2$):** In this case we set $h' = \lceil h^* \rceil$ and we want to show $v'(h')$ is not much bigger than $v(h^*)$.

First of all, since $h^* \leq h'$, we know that $n_{h^*} \leq n_{h'}$. We have

$$\sigma^2 \sum_l s_l \cdot \left(\frac{\min(h^*, s_l)}{n_{h^*} \cdot s_l}\right)^2 \geq \sigma^2 \sum_{l: h^* \geq s_l} s_l \cdot \left(\frac{\min(h^*, s_l)}{n_{h^*} \cdot s_l}\right)^2 = \sigma^2 \cdot A \cdot \left(\frac{1}{n_{h^*}}\right)^2 \geq \frac{\sigma^2}{2n_{h^*}} \geq \frac{\sigma^2}{2n_{h'}}.$$

On the other hand, we know $h^* \geq s_1 \geq 1$, and therefore $2h^* \geq \lceil h^* \rceil = h'$. So we have $\left(\frac{B \cdot h'}{\varepsilon \cdot n_{h'}}\right)^2 \leq \left(\frac{B \cdot 2h^*}{\varepsilon \cdot n_{h^*}}\right)^2 = 4 \cdot \left(\frac{B \cdot h^*}{\varepsilon \cdot n_{h^*}}\right)^2$. Putting things together, we get

$$v'(h') = \frac{\sigma^2}{n_{h'}} + 2\left(\frac{B \cdot h'}{\varepsilon \cdot n_{h'}}\right)^2 \leq 2\sigma^2 \sum_l s_l \cdot \left(\frac{\min(h^*, s_l)}{n_{h^*} \cdot s_l}\right)^2 + 4 \cdot 2\left(\frac{B \cdot h^*}{\varepsilon \cdot n_{h^*}}\right)^2 \leq 4v(h^*).$$

**Case 2 ($\alpha < 1/2$):** In this case, setting $h'$ close to $h$ might not work. We pick $\eta = 1 - \frac{1}{2(1-\alpha)}$ and $r = \lceil q \cdot \eta \rceil$, and set $h' = s_{p+r}$.

We want to show that $v'(h') \leq 4v(h^*)$. Let's start with the first parts of $v'(h')$ and $v(h^*)$. We have

$$\sigma^2 \sum_{l=1}^{m} s_l \cdot \left( \frac{\min(h^*, s_l)}{n_{h^*} \cdot s_l} \right)^2 \geq \sigma^2 \sum_{l=1}^{p+r} s_l \cdot \left( \frac{\min(h^*, s_l)}{n_{h^*} \cdot s_l} \right)^2 \geq \sigma^2 \cdot \frac{\left( \frac{\sum_{l=1}^{p+r} \min(h^*, s_l)}{n_{h^*}} \right)^2}{\sum_{l=1}^{p+r} s_l}$$

$$= \sigma^2 \cdot \frac{(\alpha + (1-\alpha) * \eta)^2}{\sum_{l=1}^{p+r} s_l} = \frac{\sigma^2}{4 \sum_{l=1}^{p+r} s_l} = \frac{v'(h')}{4}.$$

Now we consider the second parts of $v'(h')$ and $v(h^*)$. We have $\frac{h^*}{n_{h^*}} = \frac{n_{h^*} - A}{q \cdot n_{h^*}} = \frac{1-\alpha}{q}$ and

$$\frac{h'}{n_{h'}} \leq \frac{h'}{\sum_{l=p+r}^{p+q} \min(h', s_l)} = \frac{h'}{(q - r + 1)h'} \leq 1/(1 - \eta).$$

Therefore $2 \left( \frac{B \cdot h'}{\varepsilon \cdot n_{h'}} \right)^2 \leq 4 \cdot 2 \left( \frac{B \cdot h^*}{\varepsilon \cdot n_{h^*}} \right)^2$.

To sum up, we get $v'(h') \leq 4v(h^*)$. $\qquad\qquad\qquad\square$

# B  Estimating quantiles

In this section, we show that the idea of our weighted average algorithm can be also applied to estimating quantiles. We extend the PrivateQuantile algorithm of [Smi11] which is mainly based on the exponenetial mechanism [MT07] to the case when a user controls more than one data point.

Here we describe the setting of estimating quantiles. We are given $n$ samples $D = (y_1, ..., y_n)$ sampled independently from an unknown distribution $\mu$ supported on real numbers. The goal is to output the $q$-th quantile of distribution $\mu$.

As prior knowledge, we are given a base distribution $\nu$ satisfying the following assumption for some parameters $\alpha$ and $\beta$. Here $\mathbb{F}_\mu$ is the cumulative distribution function of $\mu$. This assumption can be interpreted as that elements within $\alpha$ distance in the quantile space to the $q$-th quantile of $\mu$ have probability density at least $\beta$ in the base distribution $\nu$.

**Assumption 1.** *Define set $S_{q,\alpha} = \{y \,|\, q - \alpha \leq \mathbb{F}_\mu(y) \leq q + \alpha\}$. We have $\nu(S_{q,\alpha}) \geq \beta$.*

Now we describe our algorithm for estimating quantiles. The main idea is to switch the rank function used in PrivateQuantile of [Smi11] to a weighted rank function parametrized by weights $c = (c_1, ..., c_n)$ and apply the exponential mechanism[MT07].

---
**Algorithm 4** Weighted Rank $WR_c$

---
**Input:** $y_1, \ldots, y_n \in \mathbb{R}$, $q \in (0, 1)$, base distribution $\nu$
**Parameters:** $c_1, \ldots, c_n \in \mathbb{R}_{\geq 0}$, with $\sum_{i=1}^n c_i = 1$
  1: Define the weighted rank function **weighted-rank**$(y, D) = \sum_{i=1}^n c_i \cdot \mathbb{1}\{y \geq y_i\}$.
  2: Set $W$ to be the maximum total weights of a single user, i.e. $W = \max_{j=1}^m \sum_{i \in S_j} c_i$.
  3: Sample $\tilde{y}$ with probability proportional to $\nu(y) \cdot \exp\left(-\frac{\varepsilon}{2W} \cdot |\textbf{weighted-rank}(y, D) - q|\right)$
  4: **return** $\tilde{y}$

---

**Theorem 9.** *Algorithm 4 is $\varepsilon$-DP and with probability at least $1 - 2\gamma$,*

$$|\mathbb{F}_\mu(\tilde{y}) - q| = O\left( \alpha + \sqrt{ \frac{(\max_{j=1}^m \sum_{i \in S_j} c_i)^2}{\varepsilon^2} \ln^2\left(\frac{1}{\gamma\beta}\right) + \ln\left(\frac{n}{\gamma}\right) \cdot \sum_{i=1}^n c_i^2 } \right).$$

In this section, we prove Theorem 9. Its privacy guarantee is proved in Claim 4 and its utility guarantee is proved in Corollary 1.

We first prove the privacy guarantee of Algorithm 4.

**Claim 4.** *Algorithm 4 is $\varepsilon$-DP.*

*Proof.* For any neighboring dataset $D$ and $D'$ and any $y \in \mathbb{R}$, we know the weighted rank function can be changed by at most the maximum total weight of a single user, i.e.

$$|\textbf{weighted-rank}(y, D) - \textbf{weighted-rank}(y, D')| \leq W = \max_{j=1}^{m} \sum_{i \in S_j} c_i.$$

Therefore,

$$\left| \left( -\frac{1}{2W} \cdot |\textbf{weighted-rank}(y, D) - q| \right) - \left( -\frac{1}{2W} \cdot |\textbf{weighted-rank}(y, D') - q| \right) \right| \leq \frac{1}{2}.$$

Apply Theorem 6 of [MT07], we know $WR_c$ is $\varepsilon$-DP. $\qquad\qquad\qquad\qquad\qquad\qquad\qquad\square$

We prove two claims before we proceed to Corollary 1.

**Claim 5.** *With probability $1 - \gamma$, the sampled dataset $D$ has*

$$\sup_{y \in \mathbb{R}} |F_\mu(y) - \textbf{weighted-rank}(y, D)| \leq \eta = \sqrt{\frac{1}{2} \ln\left(\frac{2n}{\gamma}\right) \cdot \sum_{i=1}^{n} c_i^2} + \max_{i \in [n]} c_i.$$

*Proof.* We start by bounding

$$\Pr_{D \sim \mu^n}\left[ \max_{i \in [n]} |F_\mu(y_i) - \textbf{weighted-rank}(y_i, D)| \leq \sqrt{\frac{1}{2} \ln\left(\frac{2n}{\gamma}\right) \cdot \sum_{i=1}^{n} c_i^2} \right].$$

By Hoeffding's inequality, we have for each $i \in [n]$,

$$\Pr_{D \sim \mu^n}\left[ |F_\mu(y_i) - \textbf{weighted-rank}(y_i, D)| \leq \sqrt{\frac{1}{2} \ln\left(\frac{2n}{\gamma}\right) \cdot \sum_{i=1}^{n} c_i^2} \right] \geq 1 - \frac{\gamma}{n}.$$

By union bound, we get

$$\Pr_{D \sim \mu^n}\left[ \max_{i \in [n]} |F_\mu(y_i) - \textbf{weighted-rank}(y_i, D)| \leq \sqrt{\frac{1}{2} \ln\left(\frac{2n}{\gamma}\right) \cdot \sum_{i=1}^{n} c_i^2} \right] \geq 1 - \gamma.$$

In the rest of the proof, it suffices to show

$$\sup_{y \in \mathbb{R}} |F_\mu(y) - \textbf{weighted-rank}(y, D)| \leq \max_{i \in [n]} |F_\mu(y_i) - \textbf{weighted-rank}(y_i, D)| + \max_{i \in [n]} c_i.$$

There are three cases:

- $y < y_1$. In this case,

$$
\begin{aligned}
&|F_\mu(y) - \textbf{weighted-rank}(y, D)| \\
=&F_\mu(y) \leq F_\mu(y_1) \\
\leq&|F_\mu(y_1) - \textbf{weighted-rank}(y_1, D)| + \textbf{weighted-rank}(y_1, D) \\
=&|F_\mu(y_1) - \textbf{weighted-rank}(y_1, D)| + c_1 \\
\leq&\max_{i \in [n]} |F_\mu(y_i) - \textbf{weighted-rank}(y_i, D)| + \max_{i \in [n]} c_i.
\end{aligned}
$$

- $y_i \leq y < y_{i+1}$ for some $i \in [n-1]$. Then we have

$$
\begin{aligned}
&\textbf{weighted-rank}(y, D) - F_\mu(y) \\
=&\textbf{weighted-rank}(y_i, D) - F_\mu(y) \\
\leq&\textbf{weighted-rank}(y_i, D) - F_\mu(y_i) \\
\leq&|F_\mu(y_i) - \textbf{weighted-rank}(y_i, D)|.
\end{aligned}
$$

and

$$F_\mu(y) - \textbf{weighted-rank}(y, D)$$
$$\leq F_\mu(y_{i+1}) - \textbf{weighted-rank}(y_i, D)$$
$$= F_\mu(y_{i+1}) - \textbf{weighted-rank}(y_{i+1}, D) + c_{i+1}$$
$$\leq |F_\mu(y_{i+1}) - \textbf{weighted-rank}(y_{i+1}, D)| + c_{i+1}.$$

In this case we also have $|F_\mu(y) - \textbf{weighted-rank}(y, D)| \leq \max_{i \in [n]} |F_\mu(y_i) - \textbf{weighted-rank}(y_i, D)| + \max_{i \in [n]} c_i.$.

- $y \geq y_n$. In this case, similarly as the first case, we can show
$$|F_\mu(y) - \textbf{weighted-rank}(y, D)| \leq |F_\mu(y_n) - \textbf{weighted-rank}(y_n, D)| + c_n.$$

$\square$

**Claim 6.** *Given the sampled dataset $D$ and the fact that the event in Claim 5 is satisfied, with probability at least $1 - \gamma$, the output $\tilde{y}$ has*

$$|\textbf{weighted-rank}(\tilde{y}, D) - q| \leq \alpha + \eta + \frac{2W}{\varepsilon} \ln\left(\frac{1}{\gamma\beta}\right).$$

*Proof.* Define two sets $\textbf{GOOD} = \{y \mid |\textbf{weighted-rank}(y, D) - q| \leq \alpha + \eta\}$ and $\textbf{BAD} = \{y \mid |\textbf{weighted-rank}(y, D) - q| \geq \alpha + \eta + \frac{2W}{\varepsilon} \ln\left(\frac{1}{\gamma\beta}\right) |\}$. We know $S_{q,\alpha} \subseteq \textbf{GOOD}$. Therefore, $\nu(\textbf{GOOD}) \geq \nu(S_{q,\alpha}) \geq \beta$. Then we have

$$\Pr[\tilde{y} \in \textbf{BAD}] \leq \frac{\Pr[\tilde{y} \in \textbf{BAD}]}{\Pr[\tilde{y} \in \textbf{GOOD}]}$$
$$\leq \exp\left(-\frac{\varepsilon}{2W} \cdot \frac{2W}{\varepsilon} \ln\left(\frac{1}{\gamma\beta}\right)\right) \frac{\nu(\textbf{BAD})}{\nu(\textbf{GOOD})}$$
$$\leq \gamma\beta \cdot \frac{1}{\beta} = \gamma.$$

$\square$

By the above two claims, we get the following corollary which bounds the distance between $\tilde{y}$ and the $q$-th quantile of $\mu$ in the quantile space.

**Corollary 1.** *With probability at least $1 - 2\gamma$,*

$$|\mathbb{F}_\mu(\tilde{y}) - q| = O\left(\alpha + \frac{W}{\varepsilon} \ln\left(\frac{1}{\gamma\beta}\right) + \sqrt{\ln\left(\frac{n}{\gamma}\right) \cdot \sum_{i=1}^{n} c_i^2}\right)$$
$$= O\left(\alpha + \sqrt{\frac{W^2}{\varepsilon^2} \ln^2\left(\frac{1}{\gamma\beta}\right) + \ln\left(\frac{n}{\gamma}\right) \cdot \sum_{i=1}^{n} c_i^2}\right).$$

## C Missing proofs of Section 4

We prove Theorem 4 in this section.

We show in the following corollary about the privacy guarantee and the weighted empirical risk of Algorithm 2. For notation convenience, define $\hat{\theta}(c, D)$ to be the minimizer of the weighted empirical risk for dataset $D$ and weights $c$, i.e. $\hat{\theta}(D) \in \arg\min_{\theta \in \mathcal{C}} \hat{L}(\theta, c, D)$.

**Corollary 2.** *Algorithm 2 is $(\varepsilon, \delta)$-DP and for weighted empirical risk, for any dataset $D = (X_1, ..., X_n)$, we have*

$$\mathbb{E}_{alg} |\hat{L}(\tilde{\theta}, c, D) - \hat{L}(\hat{\theta}(c, D), c, D)| = O\left(\frac{LB\sqrt{d}\log^2(m/\delta) \max_{j=1}^{m} \sum_{i \in S_j} c_i}{\varepsilon}\right).$$

$\mathbb{E}_{alg}$ *means that the expectation is over the randomness of the algorithm.*

*Proof.* The differential privacy guarantee of Algorithm 2 simply follows the differential privacy guarantee of Algorithm 1 in [BST14].

Notice that for each user $j$, function $l'(\cdot, M_j)$ is $(L \cdot \sum_{i \in S_j} c_i)$-Lipschitz. Applying Theorem 2.4 of [BST14] gives the bound in the corollary. □

We prove the weighted version of the uniform convergence result (Theorem 5 of [SSSS09]) to give an upper bound on the generalization error of our weighted ERM algorithm.

**Theorem 10** (Restatement of Theorem 5). *For any $\gamma > 0$, with probability at least $1 - \gamma$ over $D \sim \mu^n$, we have*

$$\sup_{\theta \in \mathcal{C}} |\hat{L}(\theta, c, D) - L_\mu(\theta)| \leq O\left(LB\sqrt{d \log(d/\gamma) \log(n) \sum_{i=1}^{n} c_i^2}\right).$$

*Proof.* By line (10) in Theorem 5 of [SSSS09], we can bound the $\ell_\infty$-covering of the class of functions $\mathcal{F} = \{X \to l(\theta, X) | \theta \in \mathcal{C}\}$:

$$\mathcal{N}(\alpha, \mathcal{F}, d_\infty) = O\left(d^2 \left(\frac{LB}{\alpha}\right)^d\right).$$

Therefore, there exists a discrete set of $\mathcal{C}' \subset \mathcal{C}$ with size $|\mathcal{C}'| = \mathcal{N}(\alpha, \mathcal{F}, d_\infty) = O\left(d^2 \left(\frac{LB}{\alpha}\right)^d\right)$, such that for any $\theta \in \mathcal{C}$, there exists a $\theta' \in \mathcal{C}'$ satisfying

$$\sup_{X \in \mathcal{X}} |l(\theta, X) - l(\theta', X)| \leq \alpha.$$

Notice that for any $\theta, \theta' \in \mathcal{C}$,

$$|\hat{L}(\theta, c, D) - L_\mu(\theta)| - |\hat{L}(\theta', c, D) - L_\mu(\theta')| \leq |\hat{L}(\theta, c, D) - \hat{L}(\theta', c, D)| + |L_\mu(\theta) - L_\mu(\theta')|$$

$$\leq \left(1 + \sum_{i=1}^{n} c_i\right) \sup_{X \in \mathcal{X}} |l(\theta, X) - l(\theta', X)|$$

$$= 2 \sup_{X \in \mathcal{X}} |l(\theta, X) - l(\theta', X)|.$$

Therefore, we can focus on the uniform convergence in set $\mathcal{C}'$:

$$\Pr\left[\sup_{\theta \in \mathcal{C}} |\hat{L}(\theta, c, D) - L_\mu(\theta)| \geq 3\alpha\right] \leq \Pr\left[\sup_{\theta \in \mathcal{C}'} |\hat{L}(\theta, c, D) - L_\mu(\theta)| \geq \alpha\right].$$

Recall $\hat{L}(\theta, c, D) = \sum_{i=1}^{n} c_i l(\theta, X_i)$ and $\mathbb{E}_{X_i \sim \mu}[l(\theta, X_i)] = L_\mu(\theta)]$. For each $\theta \in C'$, we can apply Hoeffding's inequality to show that

$$\Pr\left[|\hat{L}(\theta, c, D) - L_\mu(\theta)| \geq \alpha\right] \leq 2 \exp\left(-\frac{2\alpha^2}{\sum_{i=1}^{n} c_i^2 L^2 B^2}\right).$$

Therefore, by union bound over the set $\mathcal{C}'$, we have

$$\Pr\left[\sup_{\theta \in \mathcal{C}'} |\hat{L}(\theta, c, D) - L_\mu(\theta)| \geq \alpha\right] \leq |\mathcal{C}'| \cdot 2 \exp\left(-\frac{2\alpha^2}{\sum_{i=1}^{n} c_i^2 L^2 B^2}\right)$$

$$= O\left(d^2 \left(\frac{LB}{\alpha}\right)^d \exp\left(-\frac{2\alpha^2}{\sum_{i=1}^{n} c_i^2 L^2 B^2}\right)\right).$$

Equating the right-hand side to $\gamma$ gives the bound in the theorem. □

Finally we give the population loss of Algorithm 2.

**Corollary 3.** *If we run Algorithm 2 with weights $c = (c_1, ..., c_n)$, we can bound $\mathbb{E}_{D \sim \mu^n, alg}[L_\mu(\tilde{\theta})] - L_\mu(\theta^*)$ by*

$$O\left(LB\sqrt{d} \cdot \sqrt{\frac{\log^4(m/\delta)\left(\max_{j=1}^m \sum_{i \in S_j} c_i\right)^2}{\varepsilon^2} + \log(d)\log(n)\sum_{i=1}^n c_i^2}\right).$$

*Proof.* First notice that we can rewrite $L_\mu(\theta^*)$ as

$$L_\mu(\theta^*) = \mathbb{E}_{X \sim \mu}[l(\theta^*, X)] = \mathbb{E}_{D \sim \mu^n}\left[\sum_{i=1}^n c_i l(\theta^*, X_i)\right] = \mathbb{E}_{D \sim \mu^n}[\hat{L}(\theta^*, c, D)].$$

We prove this corollary by breaking the difference into three parts:

$$\mathbb{E}_{D \sim \mu^n, alg}[L_\mu(\tilde{\theta})] - L_\mu(\theta^*)$$

$$= \mathbb{E}_{D \sim \mu^n, alg}\Bigg[\left(\hat{L}(\hat{\theta}(c, D), c, D) - \hat{L}(\theta^*, c, D)\right) + \left(\hat{L}(\tilde{\theta}, c, D) - \hat{L}(\hat{\theta}(c, D), c, D)\right)$$

$$+ \left(L_\mu(\tilde{\theta}) - \hat{L}(\tilde{\theta}, c, D)\right)\Bigg].$$

First by the definition of $\hat{\theta}(c, D)$, we have, for any $D$,

$$\hat{L}(\hat{\theta}(c, D), c, D) - \hat{L}(\theta^*, c, D) \leq 0.$$

For the weighted empirical risk, by Corollary 2, we have

$$\mathbb{E}_{D \sim \mu^n, alg}[\hat{L}(\tilde{\theta}, c, D) - \hat{L}(\hat{\theta}(c, D), c, D)] = O\left(\frac{LB\sqrt{d}\log^2(m/\delta)\max_{j=1}^m \sum_{i \in S_j} c_i}{\varepsilon}\right).$$

For the generalization error, by Theorem 5, by picking $\gamma = 1/\sqrt{d}$, we have

$$\mathbb{E}_{D \sim \mu^n, alg}[L_\mu(\tilde{\theta}) - \hat{L}(\tilde{\theta}, c, D)] = O\left(LB\sqrt{d\log(d)\log(n)\sum_{i=1}^n c_i^2}\right)$$

To sum up, we get

$$\mathbb{E}_{D \sim \mu^n, alg}[L_\mu(\tilde{\theta})] - L_\mu(\theta^*)$$

$$\leq O\left(LB\sqrt{d}\left(\frac{\log^2(m/\delta)\max_{j=1}^m \sum_{i \in S_j} c_i}{\varepsilon} + \sqrt{\log(d)\log(n)\sum_{i=1}^n c_i^2}\right)\right)$$

$$= O\left(LB\sqrt{d} \cdot \sqrt{\frac{\log^4(m/\delta)\left(\max_{j=1}^m \sum_{i \in S_j} c_i\right)^2}{\varepsilon^2} + \log(d)\log(n)\sum_{i=1}^n c_i^2}\right).$$

$\square$

.

# D    Missing details and proofs of Section 5

## D.1    Linear regression preliminaries

In the linear regression problem, we are given $n$ data points of the form $(X_i, y_i)$, where $X_i \in \mathbb{R}^d$ and $y_i \in \mathbb{R}$ for $i = 1, \ldots, n$. The $y_i$ values are generated as

$$y_i = \beta \cdot X_i + \xi_i.$$

for a vector $\beta \in \mathbb{R}^d$ unknown to us, and random variables $\xi_i$ representing noise. These random variables are assumed to be independent, each with a mean of $0$ and a variance of $\sigma^2$. Our goal is to estimate the vector $\beta$.

A common way to solve a linear regression problem is to find $\beta$ that minimizes the *empirical loss* $\sum_{i=1}^n (\beta \cdot X_i - y_i)^2$. Let $X$ denote the $n \times d$ matrix containing $X_i$'s as its rows, and $y$ denote the $n \times 1$ vector containing $y_i$'s. We assume $X$ has rank $d$. Then, by taking derivatives and simple linear algebra, minimizing empirical risk corresponds to the following estimator:

**Definition 4** (Ordinary Least Squares Estimator (OLS)). *The ordinary least squares estimator (OLS) is*

$$\hat{\beta} = (X^T X)^{-1} X^T y.$$

The classical Gauss-Markov theorem shows that OLS is the best among all estimators satisfying two natural properties defined below.

**Definition 5** (Linear Estimators). *A linear estimator is an estimator of the form*

$$\hat{\beta} = C \cdot y,$$

*where $C$ is a $d \times n$ matrix that can depend on $X$, but not on $y$.*

**Definition 6** (Unbiased Estimators). *An estimator $\hat{\beta}$ is unbiased if for all $\beta$ and $X$, the estimates $\hat{\beta}_j$ satisfy*

$$\forall j : \quad \mathbb{E}\left[\hat{\beta}_j\right] = \beta_j.$$

**Theorem 11** (Gauss-Markov [Gau26]). *Let $\hat{\beta}$ be an estimate for $\beta$. For a vector $\lambda \in \mathbb{R}^d$, the mean squared error of this estimate at $\lambda$ is defined as $\mathbb{E}\left[(\beta \cdot \lambda - \hat{\beta} \cdot \lambda)^2\right]$.[2] Then for every $\lambda \in \mathbb{R}^d$, among all linear unbiased estimators, OLS has the lowest mean squared error at $\lambda$.*

A simple corollary of the above theorem is that among all linear unbiased estimators, OLS is the one with the minimum $\mathbb{E}\left[\sum_{j=1}^d (\beta_j - \hat{\beta}_j)^2\right]$.

We will need the following fact, which is part of the proof of the Gauss-Markov theorem.

**Fact 1.** *Let $\hat{\beta} = Cy$ be a linear estimator. $\hat{\beta}$ is an unbiased estimator if and only if $CX = I_d$, where $I_d$ is the $d \times d$ identity matrix.*

### D.2 Linear and unbiased estimators

We give some characterization of linear and unbiased estimators.

When the error random variables $\xi_i$'s (defined in Section D.1) are correlated and have $0$ mean and covariance matrix $\Omega$, the following generalized least squares estimator has been shown to be the the BLUE. And we connect generalized least squares estimator to an arbitrary linear and unbiased estimator in Claim 7.

**Definition 7** (Generalized Least Squares Estimator (GLS) [Ait34]). *The generalized least squares estimator (GLS) is*

$$\hat{\beta} = (X^T \Omega^{-1} X)^{-1} X^T \Omega^{-1} y.$$

**Claim 7.** *Any linear unbiased estimator is the GLS for some covariance matrix.*

*Proof.* Let the linear unbiased estimator be $Cy$. By Fact 1, we know that $CX = I_d$. So $\mathrm{rank}(C) \geq \mathrm{rank}(I_d) = d$ and then $\mathrm{rank}(C^T C) = \mathrm{rank}(C) \geq d$. Therefore $C^T C$ is full-rank and invertible.

Now set $\Omega = (C^T C)^{-1}$, we have

$$
\begin{aligned}
(X^T \Omega^{-1} X)^{-1} X^T \Omega^{-1} &= (X^T C^T C X)^{-1} X^T C^T C \\
&= ((CX)^T (CX))^{-1} (CX)^T C \\
&= C.
\end{aligned}
$$

So $Cy$ is the GLS for covariance matrix $(C^T C)^{-1}$. $\square$

### D.3 Missing proof of Theorem 6

**Theorem 12** (Restatement of Theorem 6). *For every $d \times n$ matrix $C$ satisfying $CX = I_d$, the algorithm $GWA_C$ is $\varepsilon$-DP and the total variance $\sum_{j=1}^{d} Var(\tilde{\beta}_j)$ of the vector $\tilde{\beta}$ produced by the algorithm $GWA_C$ can be written as:*

$$\sigma^2 \sum_{j=1}^{d} \sum_{i=1}^{n} c_{j,i}^2 + 2d \left( B\varepsilon^{-1} \cdot \max_l \sum_{j=1}^{d} \sum_{i \in S_l} |c_{j,i}| \right)^2 .$$

*This total variance is a convex function and be minimized in polynomial time.*

*Proof.* For privacy guarantee, we analyze the $\ell_1$-sensitivity of $\hat{\beta}$. If user $l$ change its input $y_i$'s for $i \in S_l$, $\hat{\beta}$ would be changed by at most $B \cdot \sum_{j=1}^{d} \sum_{i \in S_l} |c_{j,i}|$. Therefore. the $\ell_1$-sensitivity of $\hat{\beta}$ is at most $\max_{l=1}^{m} B \cdot \sum_{j=1}^{d} \sum_{i \in S_l} |c_{j,i}|$. By Theorem 1, we know that $\tilde{\beta}$ is $\varepsilon$-DP.

The calculation of the total variance is straightforward and we omitted it here.

For convexity of the total variance, we prove the claim simply by using the following facts sequentially: (1) the absolute value $f(x) = |x|$ is a convex function (2) the max of convex functions is convex (3) the square of a convex non-negative function is convex (4) the sum of convex functions is convex. $\qquad\square$

### D.4 Missing proof of Claim 3 and a different gap example

**Claim 8** (Restatement of Claim 3). *In Example 1, the minimum variance of the sample limiting algorithm is at least $1/4g^3$ and the minimum variance of the generalized weighted averaging algorithm is at most $1/g^4$.*

*Proof.* For any threshold $h \in [g]$, the sample limiting algorithm has variance

$$\max \left( \frac{g}{g^2 + h \cdot g^2}, \frac{h}{h + g^2} \right)^2 \geq \max \left( \frac{1}{2gh}, \frac{h}{2g^2} \right)^2 \geq \frac{1}{4g^3}.$$

Now we set some $C$ for the generalized weighted averaging algorithm. For data points $i$ from user $2, ..., g^2 + 1$, we set $c_{1,i}$ to be $1/g^2, 0$. For data points $i$ from user $g^2 + 3, ..., 2g^2 + 2$, we set $c_{2,i}$ to be $1/g^2$. We set other entries of $C$ to be zero. With this $C$, the generalized weighted averaging algorithm has variance

$$0 + 2d \left( \frac{B}{\varepsilon g^2} \right)^2 = \frac{1}{g^4}.$$

$\qquad\square$

**Example 3.** *$d = 2$. Let $g \geq 8$ be some integer parameter and set the number of users to be $m = g+1$. Set $\sigma = 1$ and $2d(B/\varepsilon)^2 = g$. Now consider the data points of users:*

- *User 1 has $g$ data points with $X_i = (1, 0)$.*

- *Each of user $2, ..., g + 1$ has 1 data point with $X_i = (1, 0)$ and $g - 1$ data points with $X_i = (0, 1)$.*

**Claim 9.** *In Example 3, the minimum variance of the generalized weighted averaging algorithm is at most $6/g$. For any threshold $h$, with probability at least $1/2$, the sample limiting algorithm has variance at least $g/9$.*

*Proof.* We set some $C$ for the generalized weighted averaging algorithm. For each data point $i$ with $X_i = (1, 0)$ from user $2, ..., g + 1$, we set $c_{1,i} = 1/g$. For each data point $i$ with $X_i = (0, 1)$ from user $2, ..., g + 1$, we set $c_{2,i} = 1/(g(g-1))$. We set other entries of $C$ to be zero. With this $C$, the generalized weighted averaging algorithm has variance

$$1/g + 1/(g(g-1)) + 2d \left( \frac{2B}{\varepsilon g} \right)^2 \leq \frac{6}{g}.$$

Consider the sample limiting algorithm with threshold $h$. For each of user $2, ..., g+1$, the probability that the data point with $X_i = (1,0)$ is sampled is at most $h/g$. Let $O$ be the number of sampled $X_i = (1,0)$ from user $2, ..., g+1$. We have $\mathbb{E}[O] \leq g \cdot \frac{h}{g} = h$. By Markov inequality, we know that with probability at least $1/2$, $O \leq 2h$. In this case, $\sum_{j=1}^{d} \sum_{i \in S_1^s} |c_{j,i}^s| \geq \frac{h}{h+O} \geq 1/3$ and the variance of $\tilde{\beta}^s$ is at least

$$2d(B/\varepsilon)^2 \left( \sum_{j=1}^{d} \sum_{i \in S_1^s} |c_{j,i}^s| \right)^2 \geq g/9.$$

$\square$

# E Additional material for the experimental analysis.

## E.1 Missing details of linear regression experiments

### E.1.1 Detailed description of the datasets

**Synthetic data** We generated instances of regression problems with multiple user contributions using the sklearn's package make_regression method. This method generates a random $X, y$ instance with a fix number $d_p$ ($d_n$) of predictive (non-predictive) features, as well as a controllable noise. We use $n \in [600, 3000]$ for the number of samples, $d_p = d_n = 5$, we set bias=0.0 and noise=20. To generate the number of row contributions per user we follow approximately the Zipf's (power law) distribution [AH02]. It is well known that users contribute to many systems with heavy-tailed distributions (many users have few contributions, but some have many) and the Zipf's law is well used in the literature [AH02]. We fix the number of users $u_i$ with $i$ rows, to follow the probability mass function of the Zipf's law of parameter $\alpha > 1$ (i.e. $u_i \propto \frac{1}{i^\alpha}$). We use $1 < \alpha \leq 2$ in our experiments. Then users are assigned the prescribed number of rows randomly.

**Real-world datasets** All real-world datasets are available on the UCI Machine Learning Dataset repository[3].

**drugs**[4] [GKMZ18] The dataset contains reviews of drugs by anonymous users. Each row corresponds to a drug review. The features correspond to drug characteristics (e.g., side-effects, disease treated, etc). The target to predict is the numeric rating given by the user to the drug (in range $[1, 10]$). As standard in linear regression, categorical features with $k$ values are encoded as a $k-1$ dimensional one-hot encoding dropping one category. We use the drugs as the partitions of the rows in users. We have $n = 3107$ samples, $d = 8$ features, $m = 502$ users with min 1 and max 63 samples.

**news**[5] [MT18] The dataset contains the popularity of social media posts at different times. Each row corresponds to a post. The features corresponds to the popularity of the post at intervals of 20 minutes for the first 10 hours from publication. The target to predict is the final popularity after 48 hours from publication (the range is $[0, 71]$). We use the news source as the partitions of the rows in users. We have $n = 3452$ samples, $d = 10$ features, $m = 297$ users with min 1 and max 878 samples.

### E.1.2 Detailed description of the experimental set up

For each dataset and each setting and parameter setting we run the algorithms 10 times and report mean of each metric computed. For each run of the algorithms, we use as quality measure the empirical average squared error for the prediction. We evaluate our general setting algorithm in Section 5, where the only two parameters are the $\varepsilon$ value of differential privacy and the $\sigma^2$ variance of the noise in the regression. We evaluate all algorithms using $\varepsilon = 1, 2, 3$ values. For $\sigma^2$ we plug in the variance computed by a standard ordinary least squares method (we treat it as public knowledge for simplicity of evaluation). We compare with the baseline of the **sample limiting** algorithm. For this algorithm we compare the results of the best threshold $h^*$ obtained by empirical evaluation (i.e., the one with the lowest mean empirical loss) and the method using all datasets (i.e. $h$ set to max user

Figure 3: Average squared errors for our method (solid bar) and the sample limiting method with best thresholds ('X' pattern), as well for the whole sample ('\' pattern), for the synthetic dataset with $\alpha = 1.5$ and various $\varepsilon$ parameters. – additional results

contribution). Finally, to solve the constrained optimization problem in our method we use the SCS solver of `CVXPY`[6].

### E.1.3 Additional results for the synthetic data

In Figure 3 we report additional experimental results for other settings of $\varepsilon$. The trends observed for $\varepsilon = 1$ are confirmed over the entire range. As expected, with higher $\varepsilon$ values, all methods improve their performances, but we still see that our method has lower loss. Notice also the large gap between using the best threshold and the entire dataset for the sample limiting method.

### E.2 Experiment results on logistic regression with ERM algorithms

For completeness, although our theory suggests that the weighted averaging algorithm won't outperform the sample limiting algorithm by much for ERM, we perform experiments on logistic regression using our weighted averaging algorithm and compare it to sample limiting. We use cross-entropy loss as the loss function. For a data point with feature $x$ and binary label $y$, the cross-entropy loss is defined as

$$l(\theta, (x,y)) = y \cdot \ln\left(\frac{1}{1 + e^{-\theta \cdot x}}\right) + (1 - y) \cdot \ln\left(\frac{e^{-\theta \cdot x}}{1 + e^{-\theta \cdot x}}\right).$$

We apply our weighted averaging algorithm (Algorithm 2). By Claim 1, we know the optimal weights can be characterized by a single parameter $h$, and data points from user $j$ get weight

$$\frac{1}{s_j} \cdot \frac{\min(s_j, h)}{\sum_{l=1}^{m} \min(s_l, h)}.$$

The sample limiting algorithm can also be characterized by a single parameter $h$ (needs to be an integer). For each user $j$, $\min(s_j, h)$ of its data points will be used.

We use **drugs** dataset mentioned in the linear regression experiments. We make the label binary by splitting them into rating above the median and rating below the median.

We optimize $\theta$ over $[-10, 10]^d$ and we clip the gradient of each data point at $\ell_2$ norm 1 before taking the weighted sum. For interesting comparison such that the weighted averaging algorithm and the sample limiting algorithm have reasonable performance, we set $\delta = 0.1$ and $\varepsilon = 30, 40, 50$. For each setting, we run the algorithm 50 times. We show in Table 2 the average loss of the settings. For both algorithms, $h \leq 4$ are the interesting region.

As shown in Table 2, the weighted averaging algorithm has small advantage over the sample limiting algorithm. Both the weighted averaging algorithm and the sample limiting algorithm outperform the algorithm which does not bound user contribution.

| Dataset | $\varepsilon$ | Weighted averaging | | | | Sample limiting | | | | No bounds on user contributions |
|---------|---------------|------|------|------|------|------|------|------|------|--------------------------------|
|         |               | h=1  | h=2  | h=3  | h=4  | h=1  | h=2  | h=3  | h=4  |                                |
| drugs   | 30            | 0.514 | 0.557 | 0.628 | 0.700 | 0.553 | 0.580 | 0.654 | 0.725 | 2.846 |
|         | 40            | 0.433 | 0.469 | 0.510 | 0.551 | 0.474 | 0.472 | 0.522 | 0.553 | 2.611 |
|         | 50            | 0.401 | 0.416 | 0.445 | 0.472 | 0.436 | 0.423 | 0.454 | 0.464 | 2.307 |

Table 2: Average loss for logistic regression.