[Reviews · NeurIPS 2020]

Review 1

Summary and Contributions: There is a case, when a user contributes many data points, the algorithm designer must balance between the value of the information contained in these data points, and the added noise it will have to add to the output to make it private with respect to this user. The paper follow the previous work on this problem and investigate it in the differential privacy model. They propose a weighted averaging method to smoothly bound user contributions. The main idea of this method is to set appropriate weights on data points instead of completely discarding some data points.

Strengths: 1. Motivation of their method is quite clear, which is started with mean estimation. 2. They then compare the error of such an optimal algorithm with that of the best sample limiting approach. 3. Motivated by mean estimation, they extend to ERM problem and provide the almost the same tradeoff 4. They finally study the linear regression with label privacy and investigate the same phenomenon.

Weaknesses: The trade-off is unclear, compared with the previous work. Especially, for the ERM problem in Theorem 4, we can see there is a trade-off, however, how can we get the optimal c_i? The same for linear regression with label privacy. There are also some typos. For example, it seems like Algorithm 1 needs each y_i bounded by B, which was not mentioned in the paper. Moreover, [1] also studied linear regression with label privacy, I think the author should add it to the related work. [1] Wang, Di, and Jinhui Xu. "On sparse linear regression in the local differential privacy model." International Conference on Machine Learning. 2019.

Correctness: Yes, it is correct

Clarity: Yes

Relation to Prior Work: Yes.

Reproducibility: Yes

Additional Feedback:


Review 2

Summary and Contributions: This paper considers group differential privacy in a situation where different users contribute different numbers of data points to the training data. The goal of the paper is to design algorithms for common DP tasks such as calculating a sample mean and erm so that epsilon differential privacy overall. This is achieved by reweighting data points from different users differently in a manner that minimizes an upper bound on the variance. A reweighting scheme is proposed for means as well as ERM, and a number of results are provided.

Strengths: + The problem proposed is highly practical and timely; different contributions from different number of users is indeed a major practical challenge in the deployment of differential privacy to real systems. + The algorithm proposed appears interesting and reasonable. + I believe that the algorithms and analysis proposed by the paper is novel, although there is a large body of very related work that it misses discussing and comparing with (see below).

Weaknesses: The paper suffers from two main weaknesses. 1. The exact nature of the guarantees offered is not articulated precisely and formally enough. See below for what I mean by this. 2. The paper is missing discussion of and references to a body of existing and related literature on personalized differential privacy and privacy of correlated data. Some of the issues missed by this paper is perhaps due to this reason. Expanding on 1. the algorithm for calculating the weights (and hence its privacy guarantees) depends on the number of data points contributed by each user. Thus the differential privacy offered is conditioned on the fact that these numbers are public. Again, by itself, I do not think this is a major drawback of the work, but this is an important point that should be made explicit in the paper. Another aspect that should be made more explicit is that the algorithm minimizes an upper bound on the variance of the estimator and not the variance. This non-uniform weighting introduces some bias in the estimate, which also needs to be remarked upon. In terms of 2. while the exact setting is slightly different, this work is highly related in spirit to what is by now a fairly large body of work on personalized differential privacy -- see [1] and [2] below. While in their setting each user has a different epsilon, hiding the epsilons themselves is challenging -- a challenge which is very closely related to what this paper faces -- hiding the number of data points contributed by each person. [1] Jorgensen, Zach, Ting Yu, and Graham Cormode. "Conservative or liberal? Personalized differential privacy." 2015 IEEE 31St international conference on data engineering. IEEE, 2015. [2] Alaggan, Mohammad, S├ębastien Gambs, and Anne-Marie Kermarrec. "Heterogeneous differential privacy." arXiv preprint arXiv:1504.06998 (2015).

Correctness: I believe the results of the paper are correct -- provided an explicit statement is made about the privacy guarantees offered.

Clarity: The paper needs to make explicit some of the aspects of the guarantees offered.

Relation to Prior Work: The paper misses a large body of related work on the topic of heterogeneous/personalized differential privacy. Other than the two references above, another related work in the ERM/SGD setting is [3]. There should also be a discussion of group differential privacy and why that by itself is not enough; a starting point is [4]. [3] Song, Shuang, Kamalika Chaudhuri, and Anand Sarwate. "Learning from data with heterogeneous noise using sgd." Artificial Intelligence and Statistics. 2015. [4] Kifer, Daniel, and Ashwin Machanavajjhala. "Pufferfish: A framework for mathematical privacy definitions." ACM Transactions on Database Systems (TODS) 39.1 (2014): 1-36.

Reproducibility: Yes

Additional Feedback: Overall I think this is a paper that has potential. It needs a more thorough connection to related work and a more explicit statement of the guarantees offered.


Review 3

Summary and Contributions: This paper considers the problem of bounding the weight of a single user's contributions when user-level differential privacy is desired. There is a natural tradeoff in this problem space because using more data points from some users increases the overall data used for a learning task and hence increases accuracy, but also increases sensitivity of the computation and requires more noise to be added to preserve privacy. Prior work simply bounded the number of contributed data points from each user. This work devises a weighting scheme that uses all contributed data points (which reduces variance), but down-weights the contributions of users with too many points (which reduces sensitivity). The authors apply this framework to the problems of mean estimation, ERM, estimating quantiles, and linear regression.

Strengths: I am in favor of accepting this paper. It solves a very natural and practically-relevant technical problem, provides novel results, and is cleanly presented and well-written.

Weaknesses: One issue I had with the results is that it was never stated how to computed the weights for each learning task. For some tasks, the optimization problem that the optimal weights must solve was a convex program and the authors noted it could be efficiently solved. But for some tasks the problem of finding optimal weights was not addressed. Minor comments: --In Section 3, it is never stated that the input y_is are bounded by B, and B is used in Algorithm 1 without being introduced. --Line 142: s_h should be s_m --Algorithm 2 should state how the new loss function l' is computed. --Theorem 4: why is expected excess risk the accuracy metric, rather than high-probability L2 norm as in mean estimation? This is actually partially discussed afterwards, but is never explicitly addressed. --The figures on page 8 were too small to see or interpret.

Correctness: Yes

Clarity: Yes

Relation to Prior Work: Yes

Reproducibility: Yes

Additional Feedback: Minor comment: In the proof of Lemma 3 in the supplement, Case 1 (p in M, q notin M) is incorrect. I believe this can be can be corrected by swapping the roles of p and q. As it's currently written, eta<0, which does not lead to a contradiction. ***EDIT: I have read the authors response. I appreciate them addressing the concern about computing weights. Since this was an issue for many reviewers, I suggest the authors improve discussion of this in the paper. I do not believe this is a reason for rejection, but rather something the authors can do to fend off this complaint.


Review 4

Summary and Contributions: This paper proposed three differentially private algorithms for the situations where one single user may generate multiple samples. This paper first proposes a user-level differential privacy definition, then three algorithms to weight all samples to better balance between utility and privacy guarantee. In experiments, the proposed algorithms outperform previous algorithms.

Strengths: The proposed algorithm seems flexible enough to cover lots of common scenarios such as weighted average, quantile computation, logistic regression, and the basic idea is easy to understand.

Weaknesses: Some details are missing. For example, 1. It is unknown how A in line 190 is computed, and whether it is a finite number for all ERM problems. If it can be infinite, then it would be better to have a discussion on what kind of ERM problems can benefit from the proposed mechanism. 2. it is unclear to the reviewer how optimal weights are computed. Some more details should help other researchers to reproduce results here.

Correctness: Yes

Clarity: Could be improved. Some necessary information is missing, like computation of the weights.

Relation to Prior Work: Yes.

Reproducibility: Yes

Additional Feedback:

[Author Response · NeurIPS 2020]

We thank all reviewers for their reviews.

Most reviewers raise questions about the computation of weights. We note that we only show in the paper that the
optimal weights can be computed, but it could be hard to find the closed form of the optimal weights. Here we provide
some short clarifications and we will add more explanation in the next version of the paper to make it clear.

• For mean estimation, Lemma 3 and Claim 1 on page 3 show that finding the optimal weight vector $c$ is
equivalent to minimizing a function $v(h)$ of a single parameter $h \in [s_1, s_m]$. Optimizing this function of a
single parameter can be simply done by setting the derivative to be 0 and considering locations where the
derivative is not continuous. We will add more details to help readers.

• For ERM and quantile estimation, as stated in Line 194 on page 6 of the draft, we observe that minimizing the
loss is similar to mean estimation and we can use the same method above to find the optimal weight vector.

• For linear regression with label privacy, Theorem 6 on page 7 states that the weight vector can be computed
by minimizing a convex function. We can use a zero-order convex minimization algorithm to minimize this
convex function.

Reviewer 1,3,4 point out that $B$ is not defined in Section 3. We will add the definition.

Some response to other comments:

• Reviewer 2 "We need the number of data points of users public for weight computation": You are correct. We
will make it explicit in the paper.

• Reviewer 2 "The algorithm minimizes an upper bound on the variance of the estimator ": You are correct. For
example, for mean estimation, we are minimizing the variance of the estimators in the form as Algorithm 1.

• Reviewer 2 "This non-uniform weighting introduces some bias in the estimate": In this work, we consider the
case when user data is identical distributed. Even with non-uniform weights, for example in mean estimation,
our estimator is unbiased. We agree with the reviewer that in a more complicated setting in which users have
heterogeneous sample distributions, bias will be created. We will add discussions about such bias.

• Reviewer 3 "Line 142 $s_h$ should be $s_m$": You are correct.

• Reviewer 3 "Algorith 2, defintion of new loss $l'$": $l'$ is defined in line 179. We will make the defintion of $l'$
clear in the pseudocode.

• Reviewer 3 "Why expected excess risk for Theorem 4": The expected excess risk is defined for any convex
Lipschitz loss functions. L2 norm in the mean estimation is a special case. We will add the discussion.

• Reviewer 3 "Figures on page 8 too small": We will increase the size.

• Reviewer 3 "Lemma 3 proof, Case 1": You are correct. We should swap the roles of $p$ and $q$.

Finally, for related work, we thank reviewers for providing pointers to related work in label privacy and personal-
ized/heterogeneous/group differential privacy. We will do a literature search from these pointers. We will add citations
and provide relevant discussions.

[Meta-Review · NeurIPS 2020]

The paper considers a setting in which there are multiple users, and each user contributes a possibly different amount of data. The goal is to perform a certain task while ensuring a user-level differential privacy is desired. The paper proposes an idea and method to reweigh the data so that all of the data is used but which reduces the sensitivity and thus helps in the accuracy-privacy tradeoff. The paper addresses specific problems of mean estimation, quantile estimation and logistic regression and also discusses a general ERM framework. The problem proposed is of interest to the community. The results are exciting in parts, particularly since it is a new way of addressing this non-uniform-samples problem as compared to past approaches which just bound the number of contributions from any user. The review team thus agree that this paper should be accepted. There were also various concerns raised regarding computation of the weights as well as what problems the mechanism proposed in the paper can actually solve. For instance, how can one compute the weights c's in Section 4? What about A (reviewer 4 brought up this point but wasn't addressed in the rebuttal)? More generally, what problems can the proposed mechanism solve? ^^^Please make these points very clear in the camera ready version, clearly demarcating what the paper actually solves and what it does not.^^^